# Is a Land Use Regression Model Capable of Predicting the Cleanest Route to School?

**Luca Boniardi** [1,*] , **Evi Dons** [2,3] , **Laura Campo** [4], **Martine Van Poppel** [2], **Luc Int Panis** [2,5] **and Silvia Fustinoni** [1,4]

1    EPIGET—Epidemiology, Epigenetics, and Toxicology Lab, Department of Clinical Sciences and Community Health, Università degli Studi di Milano, 20122 Milan, Italy
2    Flemish Institute for Technological Research (VITO), 2400 Mol, Belgium
3    Centre for Environmental Sciences, Hasselt University, 3590 Diepenbeek, Belgium
4    Fondazione IRCCS Ca' Granda Ospedale Maggiore Policlinico, U.O.S Tossicologia, 20122 Milan, Italy
5    Transportation Research Institute (IMOB), Hasselt University, 3590 Diepenbeek, Belgium
*    Correspondence: luca.boniardi@unimi.it

**Abstract:** Land Use Regression (LUR) modeling is a widely used technique to model the spatial variability of air pollutants in epidemiology. In this study, we explore whether a LUR model can predict home-to-school commuting exposure to black carbon (BC). During January and February 2019, 43 children walking to school were involved in a personal monitoring campaign measuring exposure to BC and tracking their home-to-school routes. At the same time, a previously developed LUR model for the study area was applied to estimate BC exposure on points along the route. Personal BC exposure varied widely with mean ± SD of 9003 ± 4864 ng/m3. The comparison between the two methods showed good agreement (Pearson's r = 0.74, Lin's Concordance Correlation Coefficient = 0.6), suggesting that LUR estimates are capable of catching differences among routes and predicting the cleanest route. However, the model tends to underestimate absolute concentrations by 29% on average. A LUR model can be useful in predicting personal exposure and can help urban planners in Milan to build a healthier city for schoolchildren by promoting less polluted home-to-school routes.

**Keywords:** air pollution; black carbon (BC); land use regression (LUR); active mobility; traffic pollution; schoolchildren; school streets

## 1. Introduction

Land Use Regression (LUR) modeling is a widely used technique to model the spatial variability of air pollutants. This approach was used in both urban and non-urban environments, usually with the aim of better predicting exposures in large epidemiological cohorts [1,2]. Especially in cities, where an important part of pollution is from traffic, identifying spatial patterns of traffic-related air pollutants (TRAP) is fundamental to enhance the accuracy of exposure assessment [3]. From personal monitoring studies, it is known that exposure while traveling is elevated mainly because of exposure peaks, and inhaled doses increase as well because of increased ventilation while traveling with active modes [4–6]. Following this, TRAP in urban areas has been one of the main health issues in the past years [7,8], and the traffic-related air pollutant Black Carbon (BC) has gained a primary role in this research field [9,10].

Personal exposure to BC in urban environments is of high health concern to many people, and even more so for schoolchildren traveling to school during rush hours [11]. In particular, we recently found that morning rush hour (MRH) during weekdays is the most critical period for exposure to TRAP

in Milan, with an average increase of BC concentration of about 1 µg/m$^3$ in both spring and winter time [12].

Moreover, in the past few years, many studies found an association between BC, changes in respiratory and cardiovascular markers, and behavioral and cognitive skills and disorders in children [13–19]. Most of the time, exposure was inferred from monitoring sites of the official Air Quality Network (AQN) or estimated by an LUR at the residential address or school site. Recently, Alvarez Pedrerol M et al. (2017) [20] linked an IQR increase of BC exposure during home-to-school commuting to a reduction in the growth of working memory in a Spanish cohort of 1234 children (aged 7–10 y). In this study, both exposure and home-to-school routes were estimated for each child, and despite this the results suggested that focusing on commuting periods can play a major role in the effort to link exposure and health outcomes.

Furthermore, LUR models can give valuable information for public health officials and urban planners to reduce population exposure to air pollution and design health-promoting cities with less polluted routes for pedestrian and cyclists. For instance, Hankey S et al. 2016 [21] combined facility-demand and LUR models to highlight exposure patterns during active travel suggesting that it should be possible to reduce exposure by ~15% after intervening on the given scenario. Recently, several studies confirmed that the health benefits linked to active travel outweigh the risks such as exposure to air pollution or accidents, suggesting that the attempt to build more walkable and bikeable urban environments is worth the effort [22–24].

Given this picture, applying LUR modeling together with GPS monitoring to the specific case of home-to-school commuting could be a valuable approach to both enhance the quality of exposure assessment in epidemiological studies and to develop new public health policies focused on the youngest generation. First, however, it is necessary to validate this approach with mobile air pollution measurements to check the effectiveness of the method.

The aim of the present study is to compare LUR model estimates and the personal BC measured concentrations of 43 schoolchildren during home-to-school commuting. Moreover, we discuss whether a LUR model can be applied to identify cleanest routes for traveling to school. The current analysis focuses on children walking to school only; commuting with other modes would imply several assumptions (such as indoor-outdoor ratios, position on the road, breathing rates) that might lead to differential misclassification [20]. This contribution is part of the "MAPS MI, Mapping Air Pollution in a School catchment area of Milan", whose aim is to study exposure to air pollution of schoolchildren in Milan, using LUR models, personal air pollution monitoring, GPS tracking, and biological monitoring with a participatory approach.

## 2. Materials and Methods

### 2.1. Study Design

The study area is about 25 km$^2$ large and is located in the north-west of Milan, which is the biggest city in the Northern Italy basin, with an estimated population of about 3.2 million residents [25]. This area represents the catchment area of an elementary school, which is attended by more than 700 children, aged 6 to 11. For the MAPS MI project, we recruited 92 schoolchildren from the school (aged 7–11) to participate in a 7-day personal monitoring campaign, performed from January 14 to February 19, 2019. During the whole period children were asked to wear a GPS device, model i-gotU GT600 (Mobile Action, Taiwan), and to fill in an activity diary. Additionally, during the last day, children wore a shoulder bag equipped with a micro-aethalometer, model AE 51 (AethLabs, San Francisco, CA, USA), for personal monitor of BC in the breathing zone (Figure 1). This included the children commuting to school in the morning between 7:30 a.m. and 8:30 a.m. Once they had reached school, the children, with the help of their accompanying person, were asked to draw their route from home-to-school on a paper map of the study area.

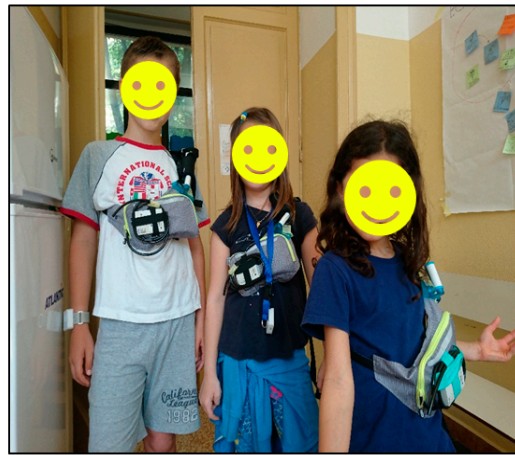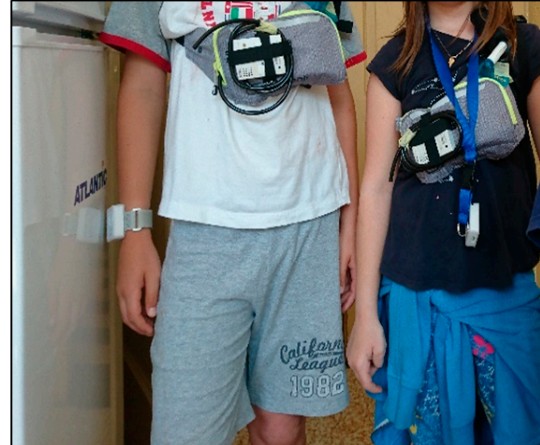

**Figure 1.** Picture of children wearing the shoulder bag equipped with a micro-aethalometer, model AE51. GPSs were worn as wristwatch or necklace.

Micro-aethalometers are optical devices that estimate the BC concentration by measuring the rate of attenuation (ATN) of a beam of light (880 nm) that passes through a T60 Teflon-coated borosilicate glass fiber filter strip over which air samples are drawn. These devices are commonly used in BC personal monitoring assessment applications thanks to: (1) their high portability; (2) the long lasting internal battery that allows about 24 h of continuous monitoring; (3) the possibility to set different flowrates and time resolutions according to the specific scenario. During our monitoring campaign, the pump flowrate was set at 100 mL/s on a 60-s time resolution. The optical measurement technique on filters may present some artifacts, the most important of which are the so-called shadowing effect, and the 880 nm beam of light multiple scattering [26]. In particular, the first relates to the increased filter loading, while the latter is linked to both filter material and aerosol composition. An overall underestimation of BC concentrations is reported in literature linked to high attenuation values and for this reason, post-processing methods are important to enhance the quality of data [27].

The project was submitted to and approved by the ethical committee of the University of Milan and the elementary school board. Before the start of the monitoring period, information and explanations about the project were given to both parents and children and permission forms were signed.

### 2.2. Data Analysis

For the present report, we focus on the route of children walking to school, resulting in a total of 43 cases (56% female). Only BC data associated with home-to-school commuting were used in this study. Raw BC data were post-processed by removing observations showing an error message, smoothing and accounting for the loading effect. In particular, we applied the Optimized Noise-reduction Algorithm (ONA) [28], and the algorithm from Virkkula et al. (2007) [29] to account for the loading effect. Furthermore, correction factors were applied to each device to correct for differences among them. These factors resulted from two intercomparison monitoring exercises carried out both before and after the personal monitoring campaign by running all devices simultaneously and next to each other for 24 h. The golden standard was micro-aethalometer model MA200-105 because (1) the device was equipped with the Dualspot© technology that automatically accounts for the loading effect; (2) the device was the most recent device calibrated by the manufacturer. The final post-processing equation is the following:

$$BC_{correct} = BC_{raw} \times (1 + ATN \times K) \times F_{IC} \qquad (1)$$

where: $BC_{correct}$ is the post-processed BC personal exposure; $BC_{raw}$ is the raw measurement performed by the micro-aethalometer; $ATN$ is the measured rate of attenuation; $K$ is the chosen Virkkula's factor; $F_{IC}$ is the intercomparison factor. The K factor represents an empirically derived constant that has the aim to correct the BC concentration underestimations that occur in the presence of filter overloading.

In particular, we used the **K** factor equal to 0.0054 calculated from a long-term monitoring campaign at the urban site of Helsinki [29]; $F_{IC}$ ranged from 0.78 to 1.00, representing the regression slopes between MA200 and AE51s that was calculated by putting together both pre- and post- intercomparison datasets.

A Land Use Regression (LUR) model was previously developed [12], and was based on a 5-week multi-site BC monitoring campaign in the school catchment area in January and February 2018. 34 sites were sampled, one week each, using micro-aethalometers, model AE51. The model used in this paper was developed by using the ESCAPE methodology [30] and selecting only weekdays BC data measured from 7 am to 9 am: a supervised forward stepwise procedure was performed using the seasonal estimated BC concentrations as the dependent variable and a pool of traffic and land use variables calculated with QGIS software as explanatory variables [31]. The best model was then selected according to the higher coefficient of determination (adjusted $R^2$) and lower Root Mean Square Error (RMSE). The final MRH LUR model was the following:

$$BC_{LUR} = 3706 \quad +(12.24 \times \textbf{\textit{TOT\_INVDist\_MRH}}) \\ +(0.001007 \times \textbf{\textit{TRAFLOAD}}\_100\_\textbf{\textit{MRH}}) \tag{2}$$

where **TOT_INVDist_MRH** is the ratio of the number of vehicles on the nearest road during MRH and the distance to that road; while **TRAFLOAD_100_MRH** is the sum of the number of vehicles on each road during MRH multiplied with the length of the roads in a circular buffer of 100 m of radius. The model $R^2$ is 0.65, the RMSE is 434 ng/m$^3$. The Leave One Out cross Validation (LOOcV) resulted in a $R^2$ of 0.51 and a RMSE of 509 ng/m$^3$.

The routes from home-to-school were drawn in a Geographical Information System (GIS) environment following the procedure below: (a) uploading GPS data and selecting only routes from home-to-school; (b) checking the paths by comparing them with the routes drawn by children and parents; (c) adjusting home-to-school routes, i.e., missing GPS data were replaced with the drawn path; (d) converting routes to 1 m equidistant points along each line. Then, for each point we estimated the BC concentration by applying the MRH LUR model. To remove possible outliers from the distribution of modeled BC values, a threshold equal to the highest seasonal BC value from the 2018 monitoring campaign +20% was set [32].

Before comparing personal exposure data and MRH LUR estimates, $BC_{LUR}$ values were rescaled as measurements, which was carried out on different days with different background concentrations. For this, we retrieved data from an urban background site of the Air Quality Network (**AQN**) of Milan (ID station 10283, via Ponzio 34/6–Pascal Città Studi), located ~6 km far from the study area. In particular, BC MRH data measured during the 2018 monitoring campaign period ($BC_{AQN\_2018}$) were compared to those measured on the specific day of the 2019 personal monitoring campaign ($BC_{AQN\_day}$). Finally, rescaled BC ($BC_{LUR\_day}$) was obtained by applying the formula below:

$$BC_{LUR\_day} = BC_{LUR} \frac{BC_{AQN\_day}}{BC_{AQN\_2018}} \tag{3}$$

R [33] was used for the statistical analysis. In particular, we performed descriptive statistical analyses including Pearson correlation analysis to compare measured and modeled exposures along the routes. We then produced a Bland-Altman plot to assess agreement between the two methods. Lin's concordance correlation coefficient was used to compare MRH LUR model estimates and the gold standard represented by the average measured personal BC concentration. The coefficient ranges from 0 to ±1, showing: (a) high concordance near +1; (b) high discordance near −1; (c) no correlation around 0.

## 3. Results

The average distance traveled by schoolchildren was 650 m (SD: 258 m), the longest route was 1403 m, while the shortest was 114 m (Table 1). The commuting trips were spread all around the school,

covering approximately the entire school catchment area. Personal BC exposure while walking to school varied widely across the monitoring campaign. The mean ± SD measured BC was 9003 ± 4864 ng/m$^3$, while 1014 ng/m$^3$ and 25,097 ng/m$^3$ respectively, were the lowest and highest values. BC values measured at the AQN background station for the monitoring period showed on an average lower concentration with a mean ± SD equal to 6635 ± 3730 ng/m$^3$.

**Table 1.** Study characteristics. 43 children who walked to school were included in the analysis. 24 children were female, 19 were male.

|  | Mean ± SD | Min–Max |
| --- | --- | --- |
| **Age (years)** | 9.1 ± 0.7 | 7–11 |
| **Distance (m)** | 650 ± 258 | 114–1403 |
| **Measured BC (ng/m$^3$)** | 9003 ± 4864 | 1014–25,097 |
| **MRH LUR BC estimate (ng/m$^3$)** | 6365 ± 3676 | 1365–12,886 |
| **MRH AQN background BC (ng/m$^3$)** | 6635 ± 3730 | 1350–14,050 |

Figure 2 shows the temporal trend of BC concentrations during the 2019 monitoring campaign. The black line represents the MRH averaged values from the AQN monitoring site and shows a marked variability during the whole period. The circles represent the average BC exposure of the schoolchildren during their home-to-school commute.

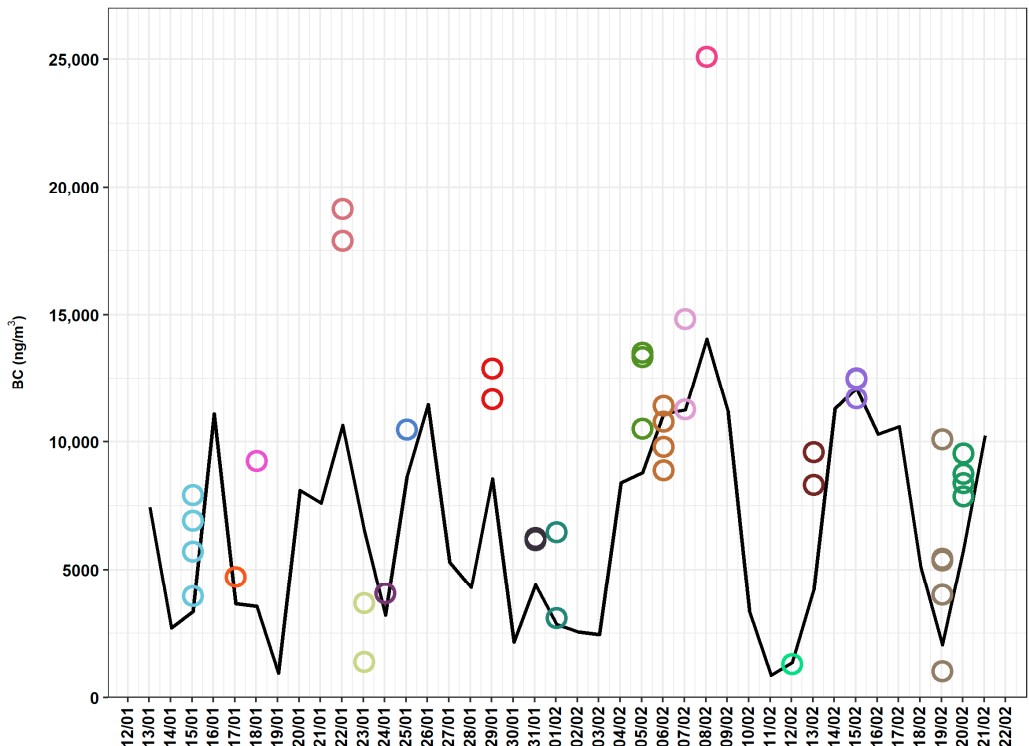

**Figure 2.** Average MRH BC concentrations measured at an urban background reference site (black line) and average home-to-school commuting personal exposure values (circles) during the 2019 personal exposure monitoring campaign. The colors of the circles correspond to the different days of monitoring.

Personal MRH LUR estimates averaged over all points along the route ranged from 1365 ng/m$^3$ to 12,886 ng/m$^3$ with mean ± SD of 6365 ± 3676 ng/m$^3$. Figure 3A shows a high correlation between average home-to-school estimates and personal BC measures (r = 0.74, *p* < 0.001). However, on average the MRH LUR model underestimated concentrations by about 29% (Table 2). The Bland-Altman plot visualizes this underestimate (blue line), and shows a slight trend in under- and over-estimating values

at the extremes (Figure 3A). Lin's Concordance Correlation Coefficient is 0.6 (95% C.I.: 0.43–0.74) and shows moderate concordance between methods.

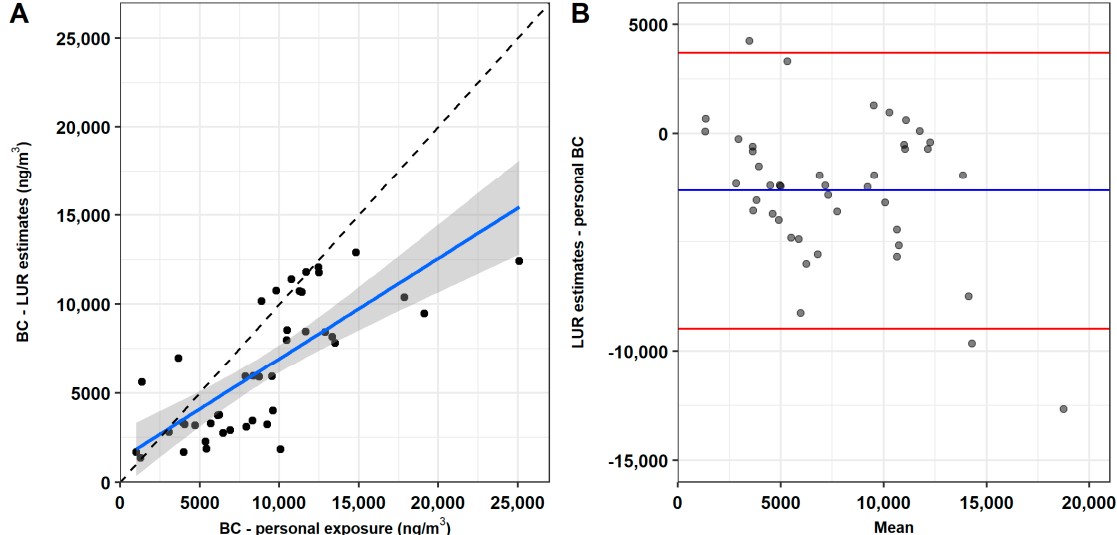

**Figure 3.** Correlation plot (**A**) with confidence interval set at 95% (shadowed area), and reference line (black dashed line). Bland-Altman plot (**B**) with average line (blue line), and ±1.96 SD lines (red lines). According to A, Pearson's correlation coefficient is 0.74. In the comparison between the two methods, Lin's Concordance Correlation Coefficient is 0.6 (95% CI: 0.43–0.74).

Table 2 and Figure 4 show five case studies comparing personal measured BC and MRH LUR BC estimates along home-to-school routes on two different days. On February 13, measured the BC of Route 2 was higher by 15% than that of Route 1. This difference is similar to the one observed between MRH LUR BC estimates for the same routes, although in absolute terms these estimates were about 30% smaller than the measured BC. On February 06, the measured BC of Route 5 was higher by 22% and 10% than for Route 4 and Route 3, respectively; the MRH LUR estimates followed a similar trend, but differences were smaller. According to the MRH LUR estimates, for this day the model tends to overestimate measured BC by about 6–15%. A visual comparison between measured BC and MRH LUR BC estimates is given in Figure 4. According to both measured BC and MRH LUR BC estimates, near the school entrance there were high BC concentrations, probably because of vehicles dropping children off in front of the school.

**Table 2.** Main information about the five case studies (see also Figure 4), such as home-to-school distance, measured and estimated BC statistics. For comparison, the mean concentration of BC at the AQN monitoring site is reported.

| Route | Day | Distance (m) | Measured BC (Mean ± SD, ng/m$^3$) | MRH LUR BC Estimate (Mean ± SD, ng/m$^3$) | MRH AQN Background BC (Mean, ng/m$^3$) |
|---|---|---|---|---|---|
| **Route 1** | 13/02/2019 | 482 | 8320 ± 1892 | 5633 ± 761 | 4200 |
| **Route 2** | 13/02/2019 | 486 | 9591 ± 2189 | 6576 ± 871 | 4200 |
| **Route 3** | 06/02/2019 | 939 | 9798 ± 2217 | 10,753 ± 2510 | 11,100 |
| **Route 4** | 06/02/2019 | 492 | 8884 ± 2125 | 10,169 ± 1954 | 11,100 |
| **Route 5** | 06/02/2019 | 1403 | 10,779 ± 4594 | 11,390 ± 2490 | 11,100 |

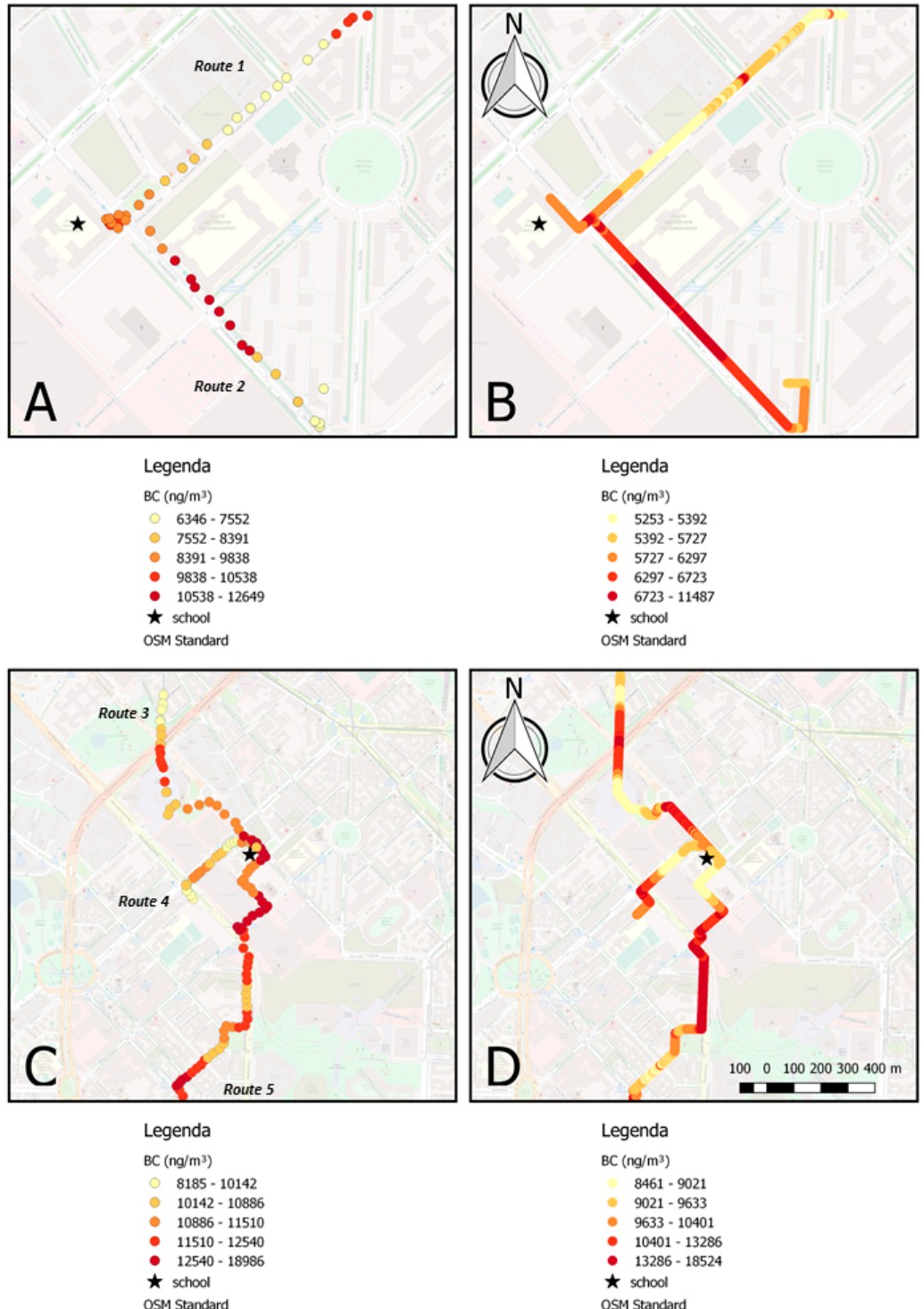

**Figure 4.** Visual comparison between measured BC and MRH LUR BC estimates of five different home-to school routes. In panel (**A**) and (**C**) there are the measured BC, while in panel (**B**) and (**D**) there are the corresponding MRH LUR BC estimates. The color is given by the quintile of the BC distribution for each day.

## 4. Discussion

To the best of our knowledge, this is the first time that a LUR model was tested to assess its reliability in estimating BC exposure during home-to-school commuting. We measured personal exposure to BC along home-to-school walking routes for 43 schoolchildren, showing considerable

variability across the entire monitoring period. Furthermore, a rescaled MRH LUR model, previously developed for the study area, was applied to the same routes to estimate personal exposure to BC. On average, the model underestimated the measured personal exposure; however, the correlation between the two methods was high. This suggests that LUR models could be successfully used to (1) highlight relative differences among routes; (2) analyze spatial patterns inside the school catchment area during home-to-school commuting; (3) predict the cleanest route from a random home in the study area to the school (Figures 3–5).

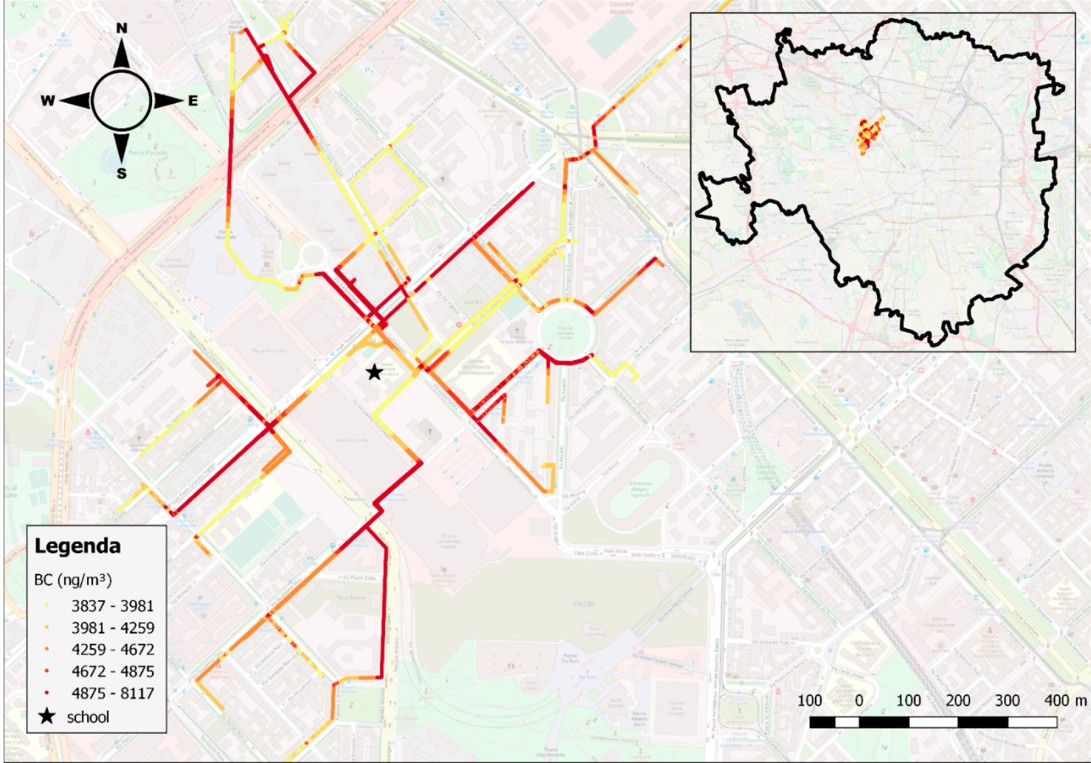

**Figure 5.** Study area and home-to-school routes. The lines were converted to equidistant points, located at 1-m distance from each other. For each point, Equation (2) was applied to estimate BC concentrations. The colors are associated to the quantile of all estimates. The inset map represents the Municipality of Milan boundaries (black line) as well as the study area (yellow-to-red lines).

This paper presents some special features and in particular: (1) schoolchildren were all monitored in a relatively short period (i.e., across January and February 2019); (2) they belonged to the same school in Milan, so the resulting study area was relatively small; (3) the MRH LUR model was developed only using MRH data collected in the school catchment area; (4) a focus on home-to-school commuting on foot. As expected, our average personal BC exposure (9003 ng/m$^3$) was higher than that reported in other studies, mainly because our data were measured during the most critical daily time-window for exposure in a congested urban environment, i.e winter MRH in Milan. For instance, Buonanno et al. (2013) [13] measured during winter an average daily BC personal exposure of 5100 ng/m$^3$ on 103 schoolchildren from Cassino, a middle town in the central part of Italy. Moreover, Nieuwenhuijsen MJ et al. (2015) [34], Paunescu et al. (2017) [35] and Cunha-Lopes I et al. (2019) [36] found an average personal BC exposure during commuting time of 2800, 3230 and 2500 ng/m$^3$ respectively in Barcelona, Lisbon and Paris. These measures were conducted on three different cohorts of children; however, they were carried out in different seasons and considering all daily trips.

Previously, other studies compared LUR estimates with personal exposure during commuting. Nieuwenhuijsen MJ et al. (2015) [34] found a lower correlation between LUR estimates at home and personal BC values during commuting (r = 0.32) than the one we found (r = 0.74). Furthermore,

Minet L et al. (2018) [37] compared BC LUR derived surfaces ($100 \times 100$ m grid) to personal exposure measures finding that the latter (median = 1764 ng/m$^3$) were underestimated by the model (median = 1469 ng/m$^3$). This result is comparable with our findings showing that short-term high exposure events are very hard to catch with a LUR approach. In contrast, the correlation between personal and modeled exposures appeared to be very low, probably due to the lack of temporal rescaling. Hankey et al. (2018) [21] applied a LUR model based on mobile monitoring to predict concentrations at the midpoint of bike path segments for a complete urban area (Minneapolis, MN, USA). Although an external validation was not provided, this analysis showed the potential of this approach for urban planning purposes.

Figure 4 shows that the spatial pattern of the personal measures and the estimates along the home-to-school routes were not always in agreement. This is probably due to short-term exposure events that cannot be captured with a LUR model. Moreover, our MRH LUR model proved to behave differently by day according to the background concentration; in particular, it was able to detect relative differences between routes, but it seemed to over- or under-estimate personal BC depending on the background concentrations. This was partially expected because of the developing procedure of the MRH LUR model and the rescaled Equation (2).

Our work has some limitations. In particular, to estimate 2019 BC concentrations we used a MRH LUR model developed using data measured by fixed monitoring sites in 2018. Then, BC estimates were rescaled only according to the concentrations at the AQN background site in 2019. This procedure has some weaknesses: it does not allow catching short-term spikes and the BC estimates are strongly connected to the behavior of the background site. Moreover, the traffic variables that we used to develop the model were only available as annual MRH estimates. Hence, we were not able to catch daily traffic conditions. Other limitations are linked to our LUR model: for instance, since we used only traffic variables, we were not able to catch the local influence of other important determinants such as the presence of restaurants or bus stops. These variables have already proved to be influential in previous works and could be useful to explain the differences between measured and modeled data [38,39]. Furthermore, we measured personal exposure to BC during home-to-school routes only on one day per child. This limited amount of data exposes our measurements to the bias linked with the temporal variation of different meteorological variables, such as wind direction and wind speed. In other words, LUR models are better suited to estimate longer-term concentrations rather than single events. Nevertheless, our results suggest that the model can be used in the study area to find relative differences between routes and to estimate the on-average cleanest walking route from an origin to a destination. Finally, in our case studies we compared only average values, without considering peak exposures or the total time of commuting. In the comparison between different routes with different distances, a cleanest route algorithm should preferably also take into account travel time. In fact, walking significantly longer to avoid polluted roads may result in a higher total uptake of pollutants; moreover, it is necessary to also consider the willingness of people to take longer routes (Anowar et al., 2017) [40].

## 5. Conclusions

Our results suggest that a LUR model is capable of predicting the cleanest walking routes to school. A further analysis of the spatial patterns of the home-to-school commute and the BC distribution in the school catchment area of the city of Milan, could provide valuable information to public health officials and urban planners. In the attempt of lowering exposure to air pollution and building a healthier city for schoolchildren, policy makers should promote less polluted home-to-school routes by, for instance, providing car-free school streets.

**Author Contributions:** Conceptualization, L.B. and E.D.; Data curation, L.B.; Formal analysis, L.B.; Funding acquisition, L.B. and S.F.; Investigation, L.B., L.C. and M.V.P.; Methodology, E.D.; Supervision, L.I.P. and S.F.; Writing–original draft, L.B., E.D., L.I.P. and S.F.; Writing–review & editing, L.B., E.D., L.I.P. and S.F.

**Funding:** This work was supported by Fondazione Cariplo [grant numbers 2017-1731].

**Acknowledgments:** We want to sincerely thank parents, operators and managers of the elementary school IC Pietro Micca, via Gattamelata 35, Milano (MI), and all the other residents of the study area that supported the project. A special thanks to all the enthusiastic involved schoolchildren.

**Conflicts of Interest:** The authors declare no conflict of interest.

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
