# Peer review of "Is a Land Use Regression Model Capable of Predicting the Cleanest Route to School?"

_environments, doi:10.3390/environments6080090_

Round 1
Reviewer 1 Report
The topic of air quality monitoring and its exposure is essential for healthy cities. The paper did a great job presenting the capabilities of Land Use regression method for short and long term study. I find the contribution of this paper significant to put forward the discussions for healthy mobility in cities, with particular consideration to the young generation like school-going children as done in this paper. Conducting the study with children and comparing the results with LUR prediction makes this paper very important for the air quality modelling community. However, I feel in terms of presentation of work, the article still needs to go through a major revision. In general, the flow of paper should be improved for schematic connections to convey the story. Please consider the following changes along with several general modifications required.
Introduction
The introduction needs further work to explain the need for the study. I understand the history of the work done, but the overall lack of connection between the aim of the research and its background is not evident in the introduction. Which gaps this study is aiming to fill or which critical aspect they want to focus is not visible. The introduction is also missing some parts which generally defines the overall structure of the paper, which makes it hard to follow the article.
Material and method
The study design section should contain the image at the place where required to convey the message. Also, the full form of the acronym MAPS Mi is repeated. The details of why aethalometers are used are not used and how is work needs more detail.
The equations use din the data analysis part is of very low resolution, makes it hard to read. Please use the proper way of creating and publishing the equations in the manuscript. Consideration to subscript and superscript need to be given. Please consider uniformity in presenting the equations and its variables in text. It is confusing to use the MRH model and MRH LUR model to talk about models please make it clear and uniform in text. The buffer used in the study was it kind of doughnut buffers or just circular buffers? Could the authors please reflect on it? Please also consider adding a table representing all the variables considered for LUR model development.
Results
This section needs work in terms of presentation of results. The text is not correctly placed. The table is overlapped with the number of line and images are far away from the place where the text is presenting its case.
It would be interesting to see the table of same route values on different dates and values. May be authors can consider adding a table for 1 or 2 routes with table 2.
I am just wondering why authors have not considered developing a LUR for each day and see the change in the variables for each temporal change (if it happens). Please shed light on that aspect too.
Please arrange the images in order than leaving pages with space.
In Figure 2, it is not clear what the colour of the circles represent?
In figure 3 and text the use of Lin’’s concordance correlation coefficient and its relevance is not well presented. Please consider improving the flow of methods adopted to explain the results better.
In Figure 4, what done + means in A+C ? can A, C present it? + is creating confusion if you are adding the result of these two or its just the presentation.
In general, the resolution of the images presented is low and hard to read. The background map needs less transparency to understand the spatial association between graphic entities better.
The page numbers of the whole manuscript are not in order.
Discussion
Along with other general corrections required the restructuring of text in this section. The general discussion and limitation subsection should be made more distinct.
Overall, the paper need major revision and detailed proofread in terms of language and logical connections between sections of article.
Reviewer 2 Report
The authors investigate the possibility of predicting BC concentrations along walking routes to school using a previously developed morning rush hour LUR model developed for this area. The modelled BC concentrations were compared to personal monitoring measurements along the routes, carried out by school children. The results showed that the modelled and measured BC concentrations correlated well, however, the LUR model tended to underestimate concentrations.
This is nice study and results suggest that LUR models can be useful to detect relative exposure differences amongst different walking routes, which is important for urban planning and to estimate cleanest walking routes.
My main concerns are related to the rescaling of the LUR estimates. As the authors briefly mention in the discussion, the rescaled values strongly depend on the background values. Looking at Figure 2 the background varies considerably across the 7 days, presumably related to different wind directions and they may not always be representative of background measurements (e.g. if downwind from city) or be the same in 2018 and 2019 for the same day. How did the background vary for the measurement days in 2018 (maybe add to Figure 2 for comparison)? Is wind data available for 2018 and 2019? Where was the background site located (show on map)? Is this a common approach?
Also, the discussion is a bit light and can be improved. In particular, it would be interesting to have a closer look at locations where the modelled and measured concentrations don’t agree and if these are associated with particular land use features that are not well captured by the LUR model.
Specific comments:
L75 Was BC only measured once? If so, how representative are these measurements of the general conditions along the walking route? What were the children measuring on the previous days?
L79 refer to Figure 1 here (instead of L81)
L97 How well did the instruments agree with each other and the golden standard?
L123 R2 or Adj. R2?
L130 Was this to remove outliers from the measured data or modelled data?
Figure 2. What do the colours represent? Just different days?
Figure 3a. It would be helpful to show the 1:1 line as well and have the same x and y axis limits.
Figure 5. What does the inset map show? City boundaries? Mention this in the figure caption.
Round 2
Reviewer 1 Report
The paper in current form could be considered for further steps. The structure of the article is not defined at the end of the introduction, which make most of the flow general. Moreover, general proofreading is needed to check the usage of punctuation marks and spellings in the paper
Author Response
Thank you for the further comments. We wrote again the last part of the introduction trying to clarify our work. We also checked and corrected punctuation marks and spellings.
Reviewer 2 Report
The reviewers have addressed my previous comments and improved the manuscript.
Author Response
Thank you once again for your comments.
Best regards